# Assessment of the impact of shared brain imaging data on the scientific literature

Michael P. Milham[1,2], R. Cameron Craddock[1,2], Jake J. Son [1], Michael Fleischmann[1], Jon Clucas [1], Helen Xu[1], Bonhwang Koo[1], Anirudh Krishnakumar[1,3], Bharat B. Biswal[4], F. Xavier Castellanos[2,5], Stan Colcombe[2], Adriana Di Martino[5], Xi-Nian Zuo [6,7,8,9] & Arno Klein[1]

Data sharing is increasingly recommended as a means of accelerating science by facilitating collaboration, transparency, and reproducibility. While few oppose data sharing philosophically, a range of barriers deter most researchers from implementing it in practice. To justify the significant effort required for sharing data, funding agencies, institutions, and investigators need clear evidence of benefit. Here, using the International Neuroimaging Data-sharing Initiative, we present a case study that provides direct evidence of the impact of open sharing on brain imaging data use and resulting peer-reviewed publications. We demonstrate that openly shared data can increase the scale of scientific studies conducted by data contributors, and can recruit scientists from a broader range of disciplines. These findings dispel the myth that scientific findings using shared data cannot be published in high-impact journals, suggest the transformative power of data sharing for accelerating science, and underscore the need for implementing data sharing universally.

[1] Center for the Developing Brain, Child Mind Institute, New York, 10022 NY, USA. [2] Center for Biomedical Imaging and Neuromodulation, Nathan S. Kline Institute for Psychiatric Research, New York, 10962 NY, USA. [3] Centre de Recherches Interdisciplinaires, INSERM U1001, Dpt Frontières du Vivant et de l'Apprendre, University Paris Descartes, Sorbonne Paris Cité, Paris, 75014, France. [4] Department of Biomedical Engineering, New Jersey Institute of Technology, Newark, 07102 NJ, USA. [5] Department of Child and Adolescent Psychiatry, Hassenfeld Children's Hospital at NYU Langone, New York, 10016 NY, USA. [6] Department of Psychology, University of Chinese Academy of Sciences (CAS), Beijing, 100049, China. [7] CAS Key Laboratory of Behavioral Science, Institute of Psychology, Beijing, 100101, China. [8] Research Center for Lifespan Development of Mind and Brain (CLIMB) and Magnetic Resonance Imaging Research Center, Institute of Psychology, Beijing, 100101, China. [9] Key Laboratory for Brain and Education Sciences, Guangxi Teachers Education University, Nanning, 530001, China. These authors contributed equally: Jake J. Son, Michael Fleischmann. Correspondence and requests for materials should be addressed to M.P.M.(email: Michael.Milham@childmind.org)

Now more than ever, the potential and actual benefits of open data sharing are being debated in the pages of premier scientific journals, funding agency communications, scientific meetings, and workshops[1–3]. Throughout these discussions an array of potential benefits are acknowledged, ranging from increased transparency of research and reproducibility of findings to decreased redundancy of effort and the generation of large-scale data repositories that can be used to achieve more appropriate sample sizes for analyses. Equally important, data sharing is commonly described as a means of facilitating collaboration across the broader scientific community.

Despite its potential, for many, the benefits of data sharing are more theoretical than practical[2,4]. The reality is that data sharing is relatively limited in many disciplines and little information on its outcomes exists[5]. In the absence of clear demonstrations of data sharing's impact, debates on the topic are dominated by formidable—albeit hypothetical—downsides. Common concerns

include loss of competitive advantage (especially for junior investigators)[6], fear of being scooped with one's own data, scientifically unsound uses of the data, and concerns that high-impact journals will not accept manuscripts that report findings generated by secondary analysis of open data sets.

To assess the tangible benefits of open data sharing, we provide a bibliometric analysis of a large brain image data-sharing initiative. The brain imaging community is a particularly valuable target for examination, as its challenges are representative of those commonly encountered in biomedical research. The high costs and workforce demands required to capture primary data limit the ability of individual labs to generate properly powered sample sizes. These obstacles are amplified when addressing more challenging (e.g., developing, aging, and clinical) populations or attempting biomarker discovery—both prerequisites for achieving clinically useful applications. Inspired by the momentum of molecular genetics,

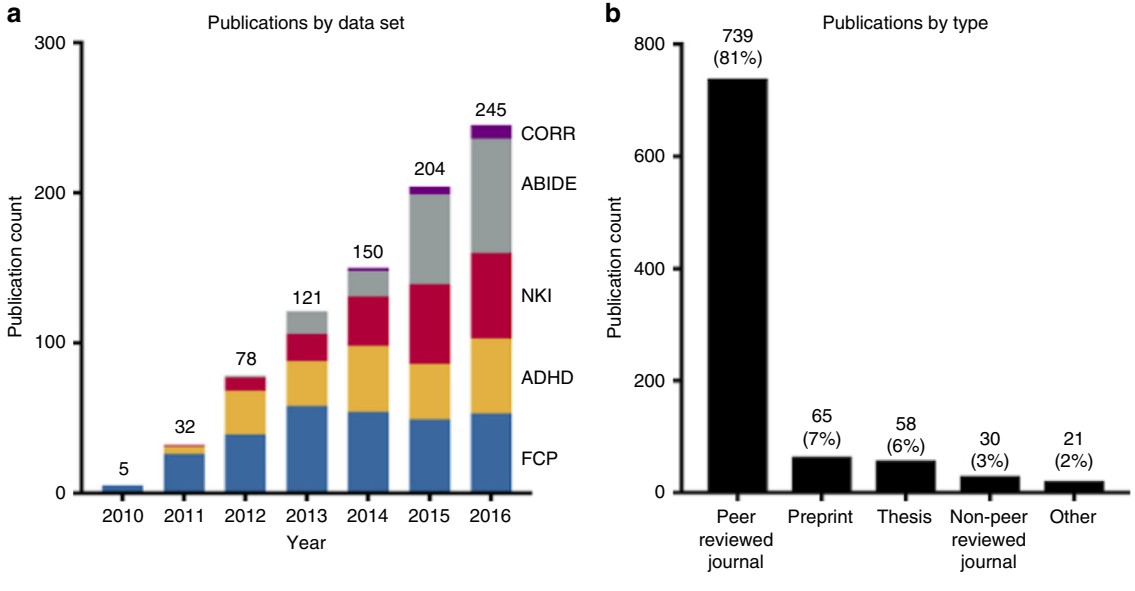

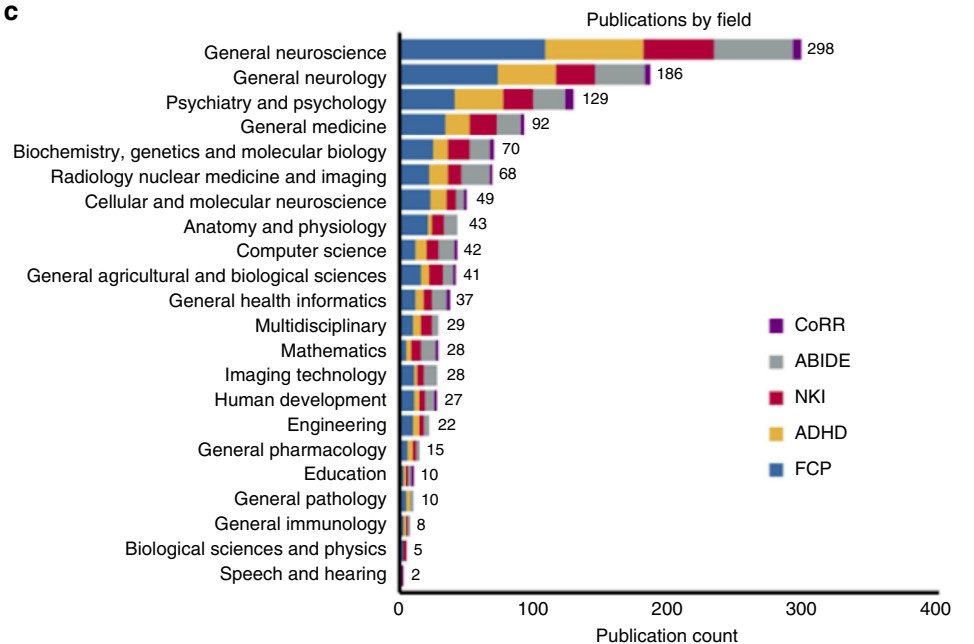

**Fig. 1** Publications that used INDI shared data. Publications sorted (**a**) by INDI data set and year, for the period of 2010–2016 (2017 is not included since that year was in progress at the time this study was conducted), (**b**) by publication type, and (**c**) by journal discipline (limited to peer-reviewed publications and based on Web of Science classifications)

the first functional neuroimaging data-sharing initiative was launched in 2000[7], though it encountered logistical challenges (e.g., lack of standardization for task-based fMRI methods) and vigorous social resistance. Since then, a range of initiatives for sharing brain imaging data have emerged (e.g., OASIS[8], ADNI[9], Human Connectome Project[10], and OpenfMRI[11]).

While some open data sharing initiatives work to aggregate and share previously collected data sets, others explicitly generate large-scale data resources for the purpose of sharing. The present work focuses on the International Neuroimaging Data-sharing Initiative (INDI)[12], as it uniquely embodies both of these models of sharing. The bibliometric measures we have employed could be easily applied to other initiatives in future work. Another distinctive aspect of INDI is its reliance on the formation of grassroots consortia as a primary vehicle for achieving its goal of aggregating and sharing previously collected data. Self-initiated and organized by scientists in the community, these consortia aggregate and share independently collected data from sites around the world. Examples of INDI-based consortia include the 1000 Functional Connectomes Project (FCP; $n = 1414$, released in December 2009)[13], the ADHD-200 ($n = 874$, released in March 2010)[14], the Autism Brain Imaging Data Exchange (ABIDE; $n = 1112$, released in August 2012)[15], and the Consortium for Reliability and Reproducibility (CoRR; $n = 1629$, released in June 2014)[16]. In the present work, we use the grassroots consortium component of the INDI model to examine the relative benefits of open data sharing versus "pay to play" models, in which only those who give data can benefit from sharing. To examine the benefits of data resources explicitly generated for the purposes of sharing, we use INDI's Nathan Kline Institute-Rockland Sample (NKI-RS) initiatives, a combination of large-scale cross-sectional and longitudinal multimodal imaging samples of brain development, maturation, and aging (ages 6.0–85.0)[17,18] (initial release in October 2010; quarterly releases ongoing, current $n = 1000+$). INDI efforts have been lauded by funding agencies, journal editors, and members of the imaging community. However, such subjective recognition does not quantify research impact. Drawing from the field of bibliometrics, we carried out a range of citation analyses[19] to quantify the impact of INDI data sets on the brain imaging and broader scientific literatures.

Our bibliometric analyses demonstrate the positive impact of openly shared data in the neuroimaging literature. In particular, they highlight the ability of openly shared data to invite the participation of scientists from a broad range of disciplines, to increase the scale of sample sizes, and to yield publications in moderate-to-high-impact journals with a frequency comparable to that of non-shared data.

## Results

**Data use.** A keyword-based search identified 1541 possible INDI-related publications as of March 22, 2017 of which 913 were determined to have used data from INDI. Figure 1a provides a non-cumulative breakdown of the 913 publications by year and initiative, revealing steady yearly increases in shared data use. Author affiliations for the 913 publications using INDI data spanned 50 countries across 6 continents, with peak affiliation densities in the United States (48.5%), China (10.7%), Germany (6.5%), and the United Kingdom (6.0%) (see http://fcon_1000.projects.nitrc.org/indi/bibliometrics/map/map.html for the world map of author affiliations generated using the Google Maps JavaScript API V3). The overwhelming majority of publications were either peer-reviewed journal articles ($n = 739$; 81%) or preprints ($n = 65$; 7%); scholarly theses had a substantial presence as well ($n = 58$ [33 doctoral; 19 master's; 4 bachelor's; 2 unspecified]), demonstrating the value of shared data for trainees and early career investigators (see Fig. 1b). As expected given the brain imaging focus of the INDI consortia, the largest proportion (45.7%) of publications were in journals focused on general neuroscience, neurology, psychiatry, and psychology. However, INDI data sets were also used in other domains (e.g., mathematics, computer science, physics, and engineering journals accounted for 6.6% of publications) (see Fig. 1c).

**Publication impact.** The impact of each of the major INDI efforts (FCP, NKI-RS, ADHD-200, ABIDE, and CoRR) on the scientific literature was quantified using an array of commonly used citation-based indices, including the h- (the number of publications with at least the same number of citations) and i10- (the number of publications with at least 10 citations) indices (see Table 1). As of March 22, 2017, the 913 publications that explicitly used INDI data had been cited 20,697 times by publications referenced in Google Scholar, with an average of 4.4 citations per article per year; h-indices for the five initiatives ranged from 7 to 52 (overall: 66) and i10-indices from 6 to 123 (overall: 295). The FCP and ADHD-200 have had the highest impact to date across various measures, though this likely reflects their older age compared to other initiatives (e.g., ABIDE and NKI-RS), which have enjoyed greater publication growth in recent years (see Fig. 1a).

To address questions about whether high-impact journals accept secondary analyses of open data, we also examined journal impact factors for publications using INDI data. The assessment of journal impact remains somewhat challenging given the growing number of indices available (e.g., impact factor, CiteScore, and altmetrics)[20], concerns about the appropriateness of judging the impact of individual articles based on measures of the impact of journals, and the potential over-reliance on these indices for promotion and tenure decisions. Nonetheless, for the purpose of citation analysis, they can be effective tools for summarizing literature-level trends. Several articles that used INDI data have been published in high-impact specialized journals (e.g., *Biological Psychiatry* and *Neuron*) and general-interest journals (e.g., *Proceedings of the National Academy of Sciences* and *Nature Communications*) (see Fig. 2a for the 15 highest-impact journals in which publications using INDI data sets have appeared based on CiteScore[20]). As shown in Fig. 2b, of all publications measured by 2015 CiteScore values, 50% were

### Table 1 Quantifying impact of INDI efforts using common publication-based indices

| Initiative | Number of papers | Total citations | Mean citations per year | Mean total citations | h-Index | h-Index (5-year) | i10-Index | i10-Index (5-year) |
|---|---|---|---|---|---|---|---|---|
| FCP | 308 | 13,147 | 7.3 ± 20.8 | 40.8 ± 140.4 | 52 | 43 | 123 | 104 |
| ADHD-200 | 210 | 2935 | 2.9 ± 5.3 | 14.5 ± 36.0 | 33 | 31 | 67 | 66 |
| ABIDE | 190 | 1875 | 2.5 ± 6.9 | 9.2 ± 28.3 | 22 | 22 | 44 | 44 |
| CoRR | 17 | 357 | 4.1 ± 7.5 | 16.3 ± 34.6 | 7 | 7 | 6 | 6 |
| NKI-RS | 188 | 2383 | 3.3 ± 5.6 | 11.9 ± 24.7 | 29 | 29 | 55 | 54 |
| Total | 913 | 20,697 | 4.4 ± 11.7 | 20.4 ± 72.9 | 66 | 58 | 295 | 274 |
| WoS | 4000 | 56,704 | 2.2 ± 3.5 | 14.2 ± 27.2 | 89 | 74 | 1506 | 1168 |

published in a journal with a CiteScore of 4.05 or higher, 25% with a CiteScore of 6.71 or higher, 5% with a score of 8.84 or higher, and 1% with a score of 12.02 or higher. Two of the three journals with the highest number of publications were *Neuro-Image* and *Human Brain Mapping*, which are among the highest-ranked field-specific brain imaging journals.

Two questions that may arise regarding our findings concerning the ability to publish in higher-impact journals using openly shared data are: (1) how would our results compare to those obtained using the same analysis applied to closed (not shared) data, and (2) would our results generalize to data-sharing initiatives other than INDI? To provide insights regarding the first question, we used PubMed to identify MRI studies of autism using the following set of keywords: (autism OR autistic OR ASD) AND (rest fMRI OR resting-state fMRI OR rest state fMRI OR Intrinsic Brain Function OR MRI OR sMRI OR functional MRI OR resting-state functional), and then programmatically vetted them—yielding 80 ABIDE and 1834 non-ABIDE studies. Of note, there were 50 ABIDE papers identified by Google Scholar that were not identified by this PubMed search. In large part, this can be explained by two factors. First, PubMed limits searches to title, abstract, and keywords; Google Scholar searches the entire document; some articles use more obscure terminology or incomplete terminology in their title and abstract. Second, a subset of papers used ABIDE data to study questions outside the autism field, such as typical brain development (using only the neurotypical data sets) or methods development. Post hoc

examination found the cumulative density curves for excluded and included papers to be highly similar. Next, we generated a cumulative density plot for studies using ABIDE data and those using closed (not shared) data. The curves we obtained were nearly identical, suggesting that papers using open data and those using closed data fared equally well with respect to the impact of the journals in which they were accepted. This result extended to the ADHD literature when the same analyses were repeated with corresponding search terms (i.e., "ADHD," "hyperactivity," "inattention" in place of "autism" and "autistic"), which revealed 38 ADHD-200 and 1986 non-ADHD-200 studies. The results for the ABIDE and ADHD-200 data sets are depicted in Fig. 2c, d, respectively.

To answer the question of whether our findings would generalize to other data-sharing initiatives, we searched for publications using HCP data. Given that this is a supplementary analysis, we used an automated search strategy that limited our search to publications in PubMed that included "Human Connectome Project" as a searchable term; as such, this analysis was meant to provide a sampling of the papers, but not to be exhaustive. Manual vetting for these results ($n = 232$) yielded 175 publications using HCP data, from which we then generated a cumulative density plot, which we found to be very similar to that obtained for INDI, suggesting that our findings generalize to other efforts. Finally, we opted to extend these findings to what would be obtained for a sampling of the entirety of the MRI imaging literature for the same time period as INDI. To

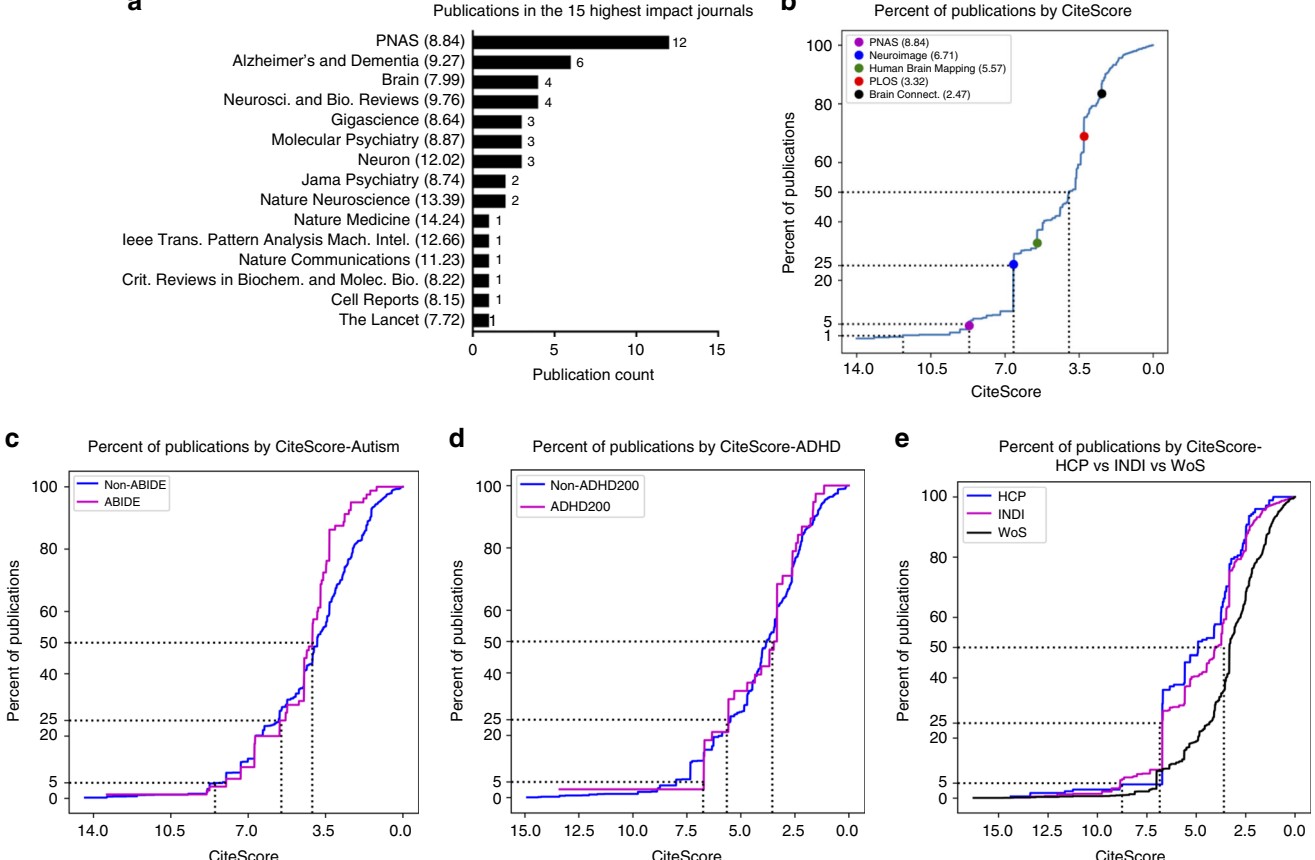

**Fig. 2** Estimation of publication impact. **a** Fifteen highest-impact journals with articles using INDI data sets (based on CiteScore, with the number of publications in each journal). **b** Cumulative density function (CDF) for CiteScores of publications that used INDI data (select journals are marked to provide reference points to help interpret CiteScores). **c** CDF comparison for MRI-based publications focused on autism that used ABIDE data versus closed data (one non-ABIDE publication of CiteScore 23.17 was not included in the figure for axis consistency). **d** CDF comparison for MRI-based publications focused on ADHD that used ADHD-200 data versus closed data. **e** CDF comparison of publications that used INDI data versus the HCP and the larger MRI brain imaging literature, as indexed by Web of Science (WoS)

accomplish this, we obtained a random sample ($n = 4000$) of papers on Web of Science containing the keywords "MRI" and "brain" within the period from 2010 to 2017 (each year was equally represented within the sample). The resulting density plot is similar to those obtained for INDI and HCP (the curves are higher for INDI and HCP over the general literature due to the greater number of publications in the higher-impact journals *NeuroImage* and *Human Brain Mapping*) (see Fig. 2e).

**Beneficiaries of sharing**. A common alternative to open data sharing is the "pay to play" model, where one must contribute data in order to gain access to data shared by others; from a consortium perspective, this means that data access is limited to members only. While such models can incentivize data sharing, they miss out on valuable analyses that researchers lacking data to contribute would perform if given the opportunity. INDI's consortium model provides a unique opportunity to compare use of shared data by contributing and non-contributing researchers. Specifically, for each initiative (FCP, ADHD-200, ABIDE, CoRR, and NKI-RS), "contributing authors" were defined as any coauthor of the announcement publication for the respective initiative. Using this definition, 90.3% of INDI-based publications were authored by research teams that did not include any data contributors. As shown in Fig. 3, the number of publications by non-contributors is rapidly increasing year to year. This differential between publications authored by contributors versus non-contributors reflects the potential missed opportunity associated with the "pay to play" model of data sharing.

The publication patterns of INDI consortia members can also be used to glean insights into the benefits of contributing data beyond inclusion as a coauthor on data announcement or descriptor papers. To accomplish this, we focused on the ADHD-200 and ABIDE consortia, as they consist of data from clinical populations, which are among the most costly to generate. For each ADHD-200 or ABIDE paper coauthored by a data contributor, we calculated the difference between the amount of data used in the publication (i.e., sample size) and the total contribution to the consortium (from coauthors on the manuscript). The median difference between publication sample size and data contribution by coauthors was 286 for ADHD-200 and 142.5 for ABIDE. Obtaining a similar increase in sample size by acquiring data from these clinical populations in a single lab

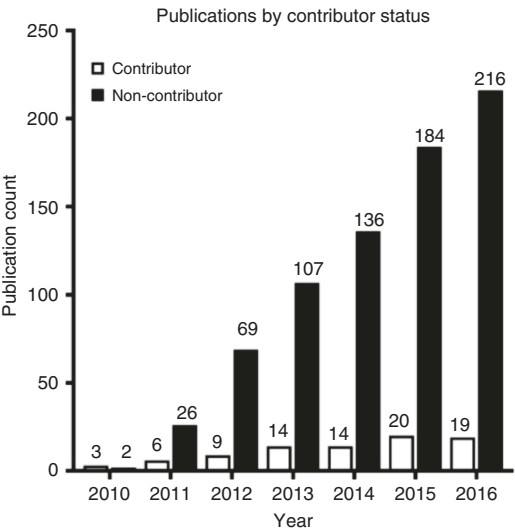

**Fig. 3** Data use by authors. Breakdown of publications by contributor status, for the period from 2010–2016 (2017 is not included since this study was conducted during that year)

would have been expensive and time consuming. Interestingly, we found that 20% of INDI-based publications from ABIDE data contributors used fewer samples than they contributed—largely reflecting more restrictive inclusion criteria related to age, sex, diagnostic phenotype, and/or image quality. Application of such criteria was made possible by the availability of shared data, and presumably enhanced the validity and reproducibility of the findings.

Another means by which shared data sets are becoming increasingly available is the resource generation model, in which data are specifically collected for the purpose of sharing (e.g., Human Connectome Project [http://www.humanconnectomeproject.org/], Brain Genomics Superstruct [http://neuroinformatics.harvard.edu/gsp/], NIH ABCD [https://addictionresearch.nih.gov/abcd-study], Child Mind Institute Healthy Brain Network [http://fcon_1000.projects.nitrc.org/indi/cmi_healthy_brain_network/], and Chinese Color Nest Project [http://zuolab.psych.ac.cn/colornest.html]). This model is advantageous in that the explicit open access intent allows researchers and funding agencies to justify investing in the creation of data resources that are notably larger in scale and broader in scope than what would typically be acquired by a single team. In INDI, the NKI-RS is an ongoing coordinated effort of three principal investigators and four NIH-funded projects dedicated to generating an open lifespan data resource for the scientific community. To date, 189 articles have been published based on NKI-RS data, 167 of which did not include the NKI-RS PIs and 76 of which were written by individuals completely outside of their publication sphere (i.e., there is no detectable relationship between the PIs and the authors of these papers based on coauthorship histories in the literature).

**Impact beyond**. It is important to note that the impact of INDI goes beyond what can be captured by the present analyses. Our searches revealed 639 publications that mentioned INDI in their text but did not use INDI data, suggesting that INDI and resultant research has impacted the thinking of authors in ways other than simply providing data. Additionally, 71 publications employed either the scripts used for the analysis of data in the initial FCP release manuscript[13] or their derivative platforms[21,22]—again highlighting utility beyond the data alone. INDI has also given rise to a set of projects sharing preprocessed data including the Preprocessed Connectomes Project (http://preprocessed-connectomes-project.org), the R-fMRI Maps Projects (http://mrirc.psych.ac.cn/RfMRIMaps), and the C3-Brain Project (http://mrirc.psych.ac.cn/3C-BrainProject); these efforts have demonstrated the feasibility of reducing barriers to data analysis (e.g., the need for domain-specific knowledge and computational resources) by sharing various processed forms of the data and establishing a quality assurance protocol.

Arguably one of the most significant forms of impact is the money saved through the reuse of data as opposed to de novo data generation for each study. Depending on the population being studied, MRI studies can vary dramatically in the costs associated with recruiting, phenotyping, and imaging. Table 2 provides an estimated cost per participant for the target populations in each of the five INDI consortia, which range from $1000 per subject for the FCP to $5000–10,000 per subject for ABIDE. Taking into account these cost estimates and the sample sizes for each of the 913 papers identified in our analyses, a conservative estimate of the cost of de novo data generation would have been $893,258,000; at the more liberal end, this estimate reaches $1,706,803,000.

Finally, it is worth noting that ready availability of the aggregate INDI resource creates an array of unique scientific

**Table 2 Quantifying the money saved through the reuse of data**

| Database | Cost/subject | Phenotyping Minimal | Phenotyping Comprehensive | Clinical Low | Population Moderate | Difficulty High | No. of publications | No. of scans/subject | $ Saved |
|---|---|---|---|---|---|---|---|---|---|
| FCP | $1000 | x | | | | | 308 | 1 | 101,003,000 |
| ADHD-200 | $2000–5000 | | | x | x | | 210 | 1 | 526,275,000 |
| NKI-RS | $3000 | | x | | | | 188 | 1 | 70,065,000 |
| ABIDE | $5000–10,000 | | | | x | x | 190 | 1 | 995,560,000 |
| CoRR | $2000 | x | | | | | 17 | 2 | 70,065,000 |

opportunities beyond increased sample size. Examples include the ability to demonstrate reproducibility of findings across independent data sets[23], to assess potential solutions for overcoming batch effects (i.e., scanner/protocol differences)[24–26], and to provide robust assessments of statistical noise[27].

## Discussion

The findings of the present work have made the theoretical benefits of open data sharing for brain imaging research tangible through citation analysis of the impact of the FCP/INDI data. Despite common misconceptions, publications using shared data are well-represented in moderate-to-high-impact journals, and were found to benefit young investigators (e.g., graduate students) as well as more senior investigators. Our findings that the scale (i.e., sample size) of studies carried out by data contributors increased through the use of data shared by others emphasized the symbiotic relationship that can emerge when investigators with common interests make their data open. Simultaneously, we demonstrated the ability to recruit the broader scientific community to address the task at hand when data are openly shared; given the various forms of expertise required to leverage technical and methodological innovations to achieve ambitions for identifying clinically useful brain-based biomarkers, the recruitment of expertise that extends beyond the teams that generated the initial data sets is essential.

Having demonstrated the impact of data sharing, a remaining challenge is to make sharing a widespread reality. Multiple advances are necessary to accomplish this change. First, there is a need for greater incentivization of both sharing and using open data sets. While funding agencies increasingly espouse mandates to share data and encourage secondary data analysis, voices vilifying openly shared data and its users[28] continue to be expressed in top journals. Second, the mechanisms for recognizing data-sharing contributions remain underspecified in, for example, grant, promotion, or tenure reviews[29]. Third, widespread data sharing requires infrastructure. To date, the storage costs of the INDI have been relatively limited, requiring about 10TB to share over 15,000 data sets. However, resource limitations must be considered as data sharing and the size of shared data sets continues to grow along with acceptance of data sharing and of data sharing mandates by funding agencies and journals. Central to these considerations will be decisions regarding the emphasis to be placed on centralized versus federated models for data storage, as the scope and scale of data sharing increases. Such decisions are non-trivial, as they entail a range of financial, logistical, and ethical questions regarding data maintenance and privacy. Fourth, there needs to be increased attention given to notification of the possibility of data sharing in the informed consent process. For studies designed with the intent to share data, whether due to their own desire or funding agency mandates, it is given that the informed consent should make clear that the data will be shared and provide details as to the nature of sharing. However, for investigators who are not committed to data sharing, a key question is whether data-sharing language

should still be included in the informed consent. This is especially important given that incentives for sharing may arise over time (e.g., data sharing supplements for funded grants, data sharing consortia relevant to the investigator's interests).

Finally, it is important to note the potential impact recent calls for harmonization can have on the value of shared data. Harmonization of protocols involves the adoption of common means for acquiring data (e.g., MRI scan sequences, experimental procedures, and phenotyping instruments), as well as the adoption of consensus standards for data quality. For the most part, discussions of harmonization in the imaging community been limited to planning large-scale multisite studies, such as the Alzheimer's Disease Neuroimaging Initiative (ADNI) and the NIH Adolescent Brain Cognitive Development (ABCD) Study. However, in recent years, there has been a growing recognition of the need to adopt common imaging protocols and phenotyping instruments as a means of improving the value of shared imaging data (see ref. [30] for a comprehensive discussion). The Human Connectome Project and ABCD Study are proving to be particularly valuable in this regard, as their imaging protocols are emerging as standards that can be adopted by investigators when there is not a clear rationale to use a custom sequence; the ABCD Study is particularly valuable, as it contains sequences for different scanners. For phenotyping, numerous efforts to develop standardized instruments that can be used to uniformly capture structured data elements across studies (i.e., common data elements) are emerging (e.g., NIH Toolbox and PhenX). In the context of INDI, ABIDE provides an excellent example of how standardization can benefit data sharing, as nearly every study in ABIDE included the Autism Diagnostic Observation Schedule (ADOS) due to long-standing efforts in the autism community to standardize phenotyping. This is in contrast to the ADHD-200, where the instrument for assessing ADHD symptoms varied from site to site. While investigators can devise strategies for pooling data across heterogeneous tools, they are inherently suboptimal.

In closing, we assert that it is the responsibility of the entire scientific ecosystem, from funding agencies to junior scientists, to accelerate the pace of progress by making data sharing the norm. While the merits and impact of data sharing are clear, it is up to all levels of science to make sharing a priority in order for its true value to be realized.

## Methods

**Identifying and classifying publications using INDI data**. We started our bibliometric analysis with a search for publications that used INDI data sets. This was a non-trivial task due to the lack of requirements for author-line recognition of INDI, a policy intended to maximize freedom of use for the data. We identified publications using a full-text search in Google Scholar; the following names and URLs were included as keywords: "fcon_1000.projects.nitrc.org," "Rockland Sample," "1000 Functional Connectomes," "International Neuroimaging Data-Sharing Initiative," "Autism Brain Imaging Data Exchange," "ADHD-200," and "Consortium for Reproducibility and Reliability." Next, we downloaded all available PDF files for manual review by a team of five research assistants, who classified each as "downloaded and used INDI subject data," "only mentions or references INDI data," "used INDI scripts but not INDI data," or "irrelevant." To facilitate this process and enable rapid review, each PDF was converted to a text file (using the Unix-based *pdftotext* shell command). Paragraphs including the keywords from

the Google Scholar search were then identified and extracted from each PDF for review in an automated fashion using regular expressions; full PDFs were available to the reviewers for verification. Classifications were determined from the consensus of two independent reviewers; conflicts were resolved by a third. During this step, research assistants also indicated the type of publication (e.g., thesis, book chapter, peer-reviewed journal article, non-peer-reviewed journal article, and preprint) for each paper.

The work reported here is largely descriptive in nature; as such, we do not include formal statistical analyses.

**Data availability**. All data and code used to generate the findings in the present work are publicly available at: https://github.com/ChildMindInstitute/Biblio_Reader/blob/165ddc56779a5e55149184a0f95b7c14874cf0c5/biblio_reader/text_tools/text_tools.py.

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

## Acknowledgements

The Child Mind Institute provides primary funding for the INDI team, with additional support provided by the Nathan S. Kline Institute for Psychiatric Research. We would like to thank the many contributors to the 1000 Functional Connectomes Project and INDI; it is their vision and contributions that have made these efforts successful. Thanks also to the many members of the INDI team over the years, especially Maarten Mennes, Quiyang Li, Dan Lurie, and David O'Connor. We thank the Neuroimaging Informatics Tools and Resources Clearinghouse (NITRC) for hosting support for INDI, as well as Amazon Web Services and the COllaborative Informatics and Neuroimaging Suite (COINS). This work was supported in part by gifts to the Child Mind Institute from Phyllis Green, Randolph Cowen, Joseph P. Healey, and the Stavros Niarchos Foundation. M.P.M. is the Phyllis Green and Randolph Cowen Scholar at the Child Mind Institute. A.D.M. received grant support from the National Institutes of Health (521MH107045); X.-N.Z. received support from the National Basic Research Program (2015CB351702), National Natural Science Foundation of China (81220108014), Beijing Municipal Science & Technology Commission (Z161100002616023 and Z171100000117012), the National R&D Infrastructure and Facility Development Program of China—"Fundamental Science Data Sharing Platform" (DKA2017−12−02−21), and Guangxi Bagui Honor Scholarship Program. A.Kr. received support from the IDEFI IIFR grant (ANR-2012-IDEFI-04). Primary funding for the NKI-RS initiatives is provided by grants from the NIH (R01MH094639, R01MH101555, R01-AG047596, and U01MH099059), as well as support from the New York State Office of Mental Health and Research Foundation for Mental Hygiene, and Child Mind Institute (1FDN2012-1).

## Author contributions

A.D.M., A.K., B.B.B., F.X.C., M.F., M.P.M., R.C.C., S.C., and X.-N.Z. envisioned the project. A.K., A.Kr., B.K., H.X., J.C., J.J.S., M.F., and M.P.M. organized and scored the data collected. Data were analyzed by A.K., J.J.S., J.C., M.F., R.C.C., M.P.M. and all authors contributed to the interpretation of findings. A.K. and M.P.M. drafted the manuscript and all authors contributed to the critical review and editing of the document.

## Additional information

**Competing interests:** The authors declare no competing interests.

