## [Peer Review File · Nature Communications]

Reviewers' comments:

Reviewer #1 (Remarks to the Author):

The manuscript by Milham et al. demonstrates the tremendous impact of the authors' open data sharing endeavors in the neuroimaging community. The study was carried by an outstanding research team that has pushed major data sharing initiatives over the last 8 years. The manuscript is well-written, figures and table are meaningful and the claims are appropriately discussed in the context of literature in the field.

The author's work had lead to remarkable studies, by INDI investigators as well as no-INDI investigators, which were published in top journals. However, since the quality of the science is at least as important as the impact factor of the journals, one wonders what high-impact studies could not have materialized timely without data sharing, and why? Also from a retrospective perspective, what is the significance of those studies today? Surely, the authors could also highlight few more high-impact studies resulting from FCP/INDI.

Strikingly, based on meticulous analyses of research publications and citations, study results show that publication success in high-impact journals is similar for data sharing and non-data sharing, at least in autism. In my opinion, the approach is highly innovative. As stated by the authors, this is important because of the common belief that scientific findings based on data sharing cannot be published in high-impact journals, which is a limiting factor for the dissemination of noble and ethically sound data sharing among researchers and institutions. However, the generalizability of the findings is unclear. Specifically, one wonders if similar results would have emerged in other disorders. It would be great if the authors could validate their findings in ADHD and/or in aging.

The "pay to play" model is unclear. Please provide a reference or give examples of initiatives using this model. I wonder if the authors refer to ENIGMA. If so, adding a comparison with ENIGMA would also be very valuable.

I appreciate the comparison with resource generation models (i.e., HCP), which is very interesting and a challenging benchmark for INDI and FCP. However, the results are unclear. The data is difficult to interpret because whereas HCP data is presented for HCP investigators and non-HCP investigators, for INDI and FCP the data are presented for all investigators. In my opinion, the authors should not discriminate among HCP and non-HCP investigators.

The extremely succinct Discussion does not properly address the low cost-to-benefit ratio of the data sharing model pushed by the authors compared to other approaches. My impression is that whereas resource generation models are tremendously expensive, INDI and FCP have been implemented with minimal additional funding from research agencies. Including funding as a weighting factor in the comparisons would help readers understand the importance of data sharing. Discussion remarks very briefly about potential benefits of future data harmonization, which I believe are key to the survival of data sharing in the advent of resource generation models. More details about this approach are needed to assess the feasibility of the proposed harmonization. Also, Discussion does not clearly express that lack of harmonization is the main limitation of current data sharing initiatives compared to resource generation initiatives. Please spell out the implications of heterogeneous MRI acquisition and incomplete phenotyping (for instance). I believe the study is important and will be well-received by the neuroscience community. I am confident the authors can address my concerns in a revision.

Reviewer #2 (Remarks to the Author):

I thoroughly enjoyed reading the manuscript, it is simple and clear yet impactful.

I could recommend acceptance as is, but I have one small request. Since we often say that sharing saves money, could you provide a 'guestimate' of the money saved with INDI. Line 186: 90% of usage is not by INDI contributors - by taking an average scan cost and the sample size used in each of these identified studies, how much money was saved? (that will speak to funders, you can even add a citation per \$ of primary vs reuse if you wanted to). Russ/Chris done this for OpenfMRI.

Cyril Pernet

Reviewer #3 (Remarks to the Author):

****Synopsis****

In this manuscript Milham and colleagues present a descriptive analysis of papers that have used shared brain imaging data. They highlight where research has been published – specifically looking at the impact factor of each journal – along with the number of citations that each paper has received. They show increasing use of the INDI datasets, and that this increase is mostly driven by non-contributors (people who did not add data to the cohort used). The paper clearly describes some of the common “selfish” concerns about sharing data and provides evidence that they are largely unwarranted.

****Compliments****

I enjoyed reading this paper and I’m glad to see some analyses that can be used to respond to the “formidable – albeit hypothetical – downsides” of data sharing that often dominate conversations around sharing data. I found the paper clear and focused.

In particular, I really liked figure 3 which shows the additional benefit to the scientific literature that sharing large datasets provides. Researchers have a lot of ideas, but there are only 24 hours in the day. The number of publications by INDI contributors is increasing modestly, but the number using the data are much larger.

Thank you for sharing the code to re-create the dataset.

****Concerns/Suggestions****

HCP analyses

I’m concerned about the analysis of the publications from the Human Connectome Project on two fronts. The first is that the analyses as they stand rather undermine the main thesis of the paper, that is that using shared data – in particular “other people’s data” has no bearing on where research will be published. The blue line of figure 2D shows there are clearly more high impact publications by the HCP investigators than by non-investigators.

The authors do address this point in their results section: “It is worth noting that the publication impact pattern for papers authored by HCP investigators did appear to have an advantage over those published by non-investigators using the same data.” Two of the three possible interpretations they provide are common reasons why researchers would **not** want to share their data: protected lead time and an advantage understanding the data. (Why make it easy for others to reuse when you can

keep publishing *and* meet your sharing requirements!)

I personally suspect that the different search strategy has a very large effect on the dataset presented. There are certainly more than 175 publications that have used HCP data, and I suspect that many of them are a) published in lower impact journals than the 28 major publications from the HCP investigators, and b) have HCP investigators named on them. My suspicion is that the papers in PubMed that include the term "Human Connectome Project" are a non-representative sample of the actual papers that use HCP data.

I'm not sure exactly what I recommend on this front, I understand from the methods section that it was a lot of work for the 5 research assistants to track all the publications using INDI datasets, so it may not be possible to find all the publications that have used HCP data. I would be interested to hear the authors' response to my concern though, as they may be able to justify the sampling strategy better than is currently described in the paper.

Eitherway, I think it would be helpful to the reader to expand the discussion of the discrepancy between the HCP investigators and the INDI contributors. These two initiatives are very different in their approach to sharing and while I agree that the non-contributor users generalise nicely, the benefits to the organisers may be worth drawing attention as many of the readers will find (or have already found) themselves on the receiving end of a "data sharing mandate" from their funding body and there are major decisions to make regarding how accessible to make the data (even when officially available).

Statistics

This isn't a big deal, but there are no statistics performed in the paper. I don't think there necessarily need to be, but I wonder if a sentence acknowledging that this is a descriptive study could be added to the methods section to make this clear.

Comparative indices of impact

As I was reading the paragraph on publication impact in the results section (line 125) I wondered how these citation and impact metrics would compare to a randomly selected comparative sample. For example, I might hypothesise that these studies are more highly cited than others due to the larger (on average) sample sizes. Comparing these measures for the non-ABIDE studies (N=1834) or for the HCP studies (N=175) could be an easy analysis, but I'd be most interested to know how the measures compare to a sample of all human neuroimaging studies.

Informed consent

The paper does not address one of the major barriers to sharing data, that of the requirement to obtain informed consent before data can be made accessible to users. I don't think this manuscript needs to also add in a lot of detail about this barrier, but I wonder if it could be acknowledged somewhere in the paper to make this a slightly more complete resource for readers.

Measuring impact

The authors mention that "...the assessment of journal impact remains somewhat controversial given the growing number of indices available (e.g., impact factor, CiteScore, altmetrics)" but I'd like to point out that it is not simple the number of indices that make JIF a controversial measure. A good discussion of their limitations can be found in Lariviere et al, BioRxiv, 2016, <https://doi.org/10.1101/062109>. I'd like to see more acknowledgement of these limitations in the

introduction, along with an acknowledgement that they *are* used to assess tenure, promotion, job offers etc (and therefore the current study is worth conducting).

As a minor note, please describe how the h and i10 indices are calculated, as these terms are not defined in the text.

Data sharing

As I mentioned above, thank you for sharing the code required for the analyses. As the data will change between when the study was conducted and when the reader learns of it, it would be nice to have an archive of the dataset used to make the plots/tables in the manuscript, eg by uploading to an institutional repository or zenodo.

Kirstie Whitaker

Nature Communications Response to Reviewers

Reviewer #1 (Remarks to the Author)

The manuscript by Milham et al. demonstrates the tremendous impact of the authors' open data sharing endeavors in the neuroimaging community. The study was carried by an outstanding research team that has pushed major data sharing initiatives over the last 8 years. The manuscript is well-written, figures and table are meaningful and the claims are appropriately discussed in the context of literature in the field.

REVIEWER: The author's work had lead to remarkable studies, by INDI investigators as well as no-INDI investigators, which were published in top journals. However, since the quality of the science is at least as important as the impact factor of the journals, one wonders what high-impact studies could not have materialized timely without data sharing, and why? Also from a retrospective perspective, what is the significance of those studies today? Surely, the authors could also highlight few more high-impact studies resulting from FCP/INDI.

RESPONSE: We appreciate the reviewer's comments and suggestions regarding this point. In response, we have included an increased emphasis on some of the unique efforts that could not have been carried out in a timely manner without open data sharing. We believe that combined with the financial analysis suggested by Reviewer 2, this addition makes the impact of sharing more clear.

“Impact Beyond. It is important to note that the impact of INDI goes beyond what can be captured by the present analyses. Our searches revealed 639 publications that mentioned INDI in their text but did not use INDI data, suggesting that INDI and resultant research has impacted the thinking of authors in ways other than simply providing data. Additionally, 71 publications employed either the scripts used for the analysis of data in the initial FCP release manuscript¹³ or their derivative platforms (^{21,22}) - again highlighting utility beyond the data alone. INDI has also given rise to a set of projects sharing preprocessed data including the Preprocessed Connectomes Project (<http://preprocessed-connectomes-project.org>), the R-fMRI Maps Projects (<http://mrirc.psych.ac.cn/RfMRIMaps>) and the C3-Brain Project (<http://mrirc.psych.ac.cn/3C-BrainProject>); these efforts have demonstrated the feasibility of reducing barriers to data analysis (e.g., the need for domain-specific knowledge and computational resources) by sharing various processed forms of the data and establishing a quality assurance protocol.

Arguably one of the most significant forms of impact is the money saved through the reuse of data as opposed to de novo data generation for each study. Depending on the population being studied, MRI studies can vary dramatically in the costs associated with recruiting, phenotyping and imaging. Supplementary Table S1 provides an estimated cost per participant for the target populations in each of the five INDI consortia, which range from \$1,000 per subject for the FCP to \$5,000-10,000 per subject for ABIDE. Taking into account these cost estimates and the sample sizes for each of the 913 papers identified in our analyses, a conservative estimate of the cost of de novo data generation would have been \$893,258,000; at the more liberal end, this estimate reaches \$1,706,803,000.

Finally, it is worth noting that ready availability of the aggregate INDI resource creates an

array of unique scientific opportunities beyond increased sample size. Examples include the ability to demonstrate reproducibility of findings across independent datasets (e.g., ²³), to assess potential solutions for overcoming batch effects (i.e., scanner/protocol differences) (e.g., ²⁴⁻²⁶), and to provide robust assessments of statistical noise (e.g., ²⁷).”

REVIEWER: Strikingly, based on meticulous analyses of research publications and citations, study results show that publication success in high-impact journals is similar for data sharing and non-data sharing, at least in autism. In my opinion, the approach is highly innovative. As stated by the authors, this is important because of the common belief that scientific findings based on data sharing cannot be published in high-impact journals, which is a limiting factor for the dissemination of noble and ethically sound data sharing among researchers and institutions. However, the generalizability of the findings is unclear. Specifically, one wonders if similar results would have emerged in other disorders. It would be great if the authors could validate their findings in ADHD and/or in ageing.

RESPONSE: We have extended our journal impact factor analyses to examine the ADHD-200 as well. As expected, the observed pattern was very similar to that observed for ABIDE. We have updated our figure to include analyses comparing the cumulative density functions for ADHD-200 and non-ADHD-200 studies, and the resulting figure shows almost identical curves. We have updated the text in the legend for Figure 2 according, as well as in the document as follows:

“This result extended to the ADHD literature when the same analyses were repeated with corresponding search terms (i.e., “ADHD”, “hyperactivity”, “inattention” in place of “autism” and “autistic”), which revealed 38 ADHD-200 and 1,986 non-ADHD-200 studies. The results for the ABIDE and ADHD-200 data are depicted in Figure 2c and 2d respectively.”

REVIEWER: The “pay to play” model is unclear. Please provide a reference or give examples of initiatives using this model. I wonder if the authors refer to ENIGMA. If so, adding a comparison with ENIGMA would also be very valuable.

RESPONSE: We have clarified our definition of the pay to play model as follows: “A common alternative to open data sharing is the “pay to play” model, where one must contribute data in order to gain access to the data shared by others; from a consortium perspective, this means that data access is limited to members only.”

This definition applies to any consortium that limits access to those laboratories that contributed data; we are opting not to name any examples to avoid negative connotations.

REVIEWER: I appreciate the comparison with resource generation models (i.e., HCP), which is very interesting and a challenging benchmark for INDI and FCP. However, the results are unclear. The data is difficult to interpret because whereas HCP data is presented for HCP investigators and non-HCP investigators, for INDI and FCP the data are presented for all investigators. In my opinion, the authors should not discriminate among HCP and non-HCP investigators.

RESPONSE: We appreciate the reviewer's concerns about separating out HCP and non-HCP investigators, which were echoed by Reviewer #3. As suggested, we revised our manuscript to no longer discriminate among HCP and non-HCP investigators. The text has been revised as follows:

"To answer the question of whether our findings would generalize to other data sharing initiatives, we searched for publications using HCP data. Given that this is a supplementary analysis, we used an automated search strategy that limited our search to publications in PubMed that included "Human Connectome Project" as a searchable term; as such, this analysis was meant to provide a sampling of the papers, but not to be exhaustive. Manual vetting for these results (n= 232) yielded 175 publications using HCP data, from which we then generated a cumulative density plot, which we found to be very similar to that obtained for INDI, suggesting that our findings generalize to other efforts. Finally, we opted to extend these findings to what would be obtained for a sampling of the entirety of the MRI imaging literature for the same time period as INDI. To accomplish this, we obtained a random sample (n = 4,000) of papers on Web of Science containing the keywords "MRI" and "brain" within the period from 2010 to 2017 (each year was equally represented within the sample). The resulting density plot is similar to those obtained for INDI and HCP (the curves are higher for INDI and HCP over the general literature due to the greater number of publications in the higher-impact journals NeuroImage and Human Brain Mapping) (see Figure 2e)."

REVIEWER: The extremely succinct Discussion does not properly address the low cost-to-benefit ratio of the data sharing model pushed by the authors compared to other approaches. My impression is that whereas resource generation models are tremendously expensive, INDI and FCP have been implemented with minimal additional funding from research agencies. Including funding as a weighting factor in the comparisons would help readers understand the importance of data sharing. Discussion remarks very briefly about potential benefits of future data harmonization, which I believe are key to the survival of data sharing in the advent of resource generation models. More details about this approach are needed to assess the feasibility of the proposed harmonization. Also, Discussion does not clearly express that lack of harmonization is the main limitation of current data sharing initiatives compared to resource generation initiatives.

Please spell out the implications of heterogeneous MRI acquisition and incomplete phenotyping (for instance). I believe the study is important and will be well-received by the neuroscience community. I am confident the authors can address my concerns in a revision.

RESPONSE: As suggested by the reviewer, we have expanded the discussion to address these points as follows:

“Finally, it is important to note the potential impact recent calls for harmonization can have on the value of shared data. Harmonization of protocols involves the adoption of common means for acquiring data (e.g., MRI scan sequences, experimental procedures, phenotyping instruments), as well as the adoption of consensus standards for data quality. For the most part, discussions of harmonization in the imaging community been limited to planning large-scale multisite studies, such as the Alzheimer’s Disease Neuroimaging Initiative (ADNI) and the NIH Adolescent Brain Cognitive Development (ABCD) Study. However, in recent years there has been a growing recognition of the need to adopt common imaging protocols and phenotyping instruments as a means of improving the value of shared imaging data (see³⁰ for a comprehensive discussion). The Human Connectome Project and ABCD Study are proving to be particularly valuable in this regard, as their imaging protocols are emerging as standards that can be adopted by investigators when there is not a clear rationale to use a custom sequence; the ABCD Study is particularly valuable, as it contains sequences for different scanners. For phenotyping, numerous efforts to develop standardized instruments that can be used to uniformly capture structured data elements across studies (i.e., common data elements) are emerging (e.g., NIH Toolbox, PhenX). In the context of INDI, ABIDE provides an excellent example of how standardization can benefit data sharing, as nearly every study in ABIDE included the Autism Diagnostic Observation Schedule (ADOS) due to long-standing efforts in the autism community to standardize phenotyping. This is in contrast to the ADHD-200, where the instrument for assessing ADHD symptoms varied from site to site. While investigators can devise strategies for pooling data across heterogeneous tools, they are inherently suboptimal.

In closing, we assert that it is the responsibility of the entire scientific ecosystem, from funding agencies to junior scientists, to accelerate the pace of progress by making data sharing the norm. While the merits and impact of data sharing are clear, it is up to all levels of science to make sharing a priority in order for its true value to be realized.”

Reviewer #2 (Remarks to the Author):

I thoroughly enjoyed reading the manuscript, it is simple and clear yet impactful.

REVIEWER: I could recommend acceptance as is, but I have one small request. Since we often say that sharing saves money, could you provide a 'guestimate' of the money saved with INDI. Line 186: 90% of usage is not by INDI contributors - by taking an average scan cost and the sample size used in each of these identified studies, how much money was saved? (that will speak to funders, you can even add a citation per \$ of primary vs reusage if you wanted to). Russ/Chris done this for OpenfMRI.

Cyril Pernet

RESPONSE: We appreciate the reviewer's suggestion. To best estimate the savings achieved through INDI, we took the sample size for each manuscript and multiplied it by the estimated cost of the specific INDI datasets used. The cost table is provided below:

Database	Cost / Subject	Phenotyping		Clinical Population Difficulty			# Publications	# Scans/Subject	\$ Saved
		Minimal	Comprehensive	Low	Moderate	High			
FCP	\$1,000	✓					308	1	101,003,000
ADHD-200	\$2000-5000			✓	✓		210	1	526,275,000
NKI-RS	\$3,000		✓				188	1	70,065,000
ABIDE	\$5000-10000				✓	✓	190	1	995,560,000
CoRR	\$2,000	✓					17	2	70,065,000

The following text was added to discuss these findings:

“Arguably, one of the most ostensible forms of impact is the money saved through the reuse of data as opposed to de novo data generation for each study. Depending on the population being studied, MRI studies can vary dramatically in the costs associated with recruiting, phenotyping and imaging a population of interest. Supplementary table S1 provides an estimated cost per participant for the target populations in each of the five INDI consortia, which ranged from \$1000 per subject for the FCP to \$5000-10,000 per subject for ABIDE. Taking into account these cost estimates and the sample sizes for each of the 913 papers identified in our analyses, a conservative estimate of the cost of de novo data generation would have been \$893,258,000; at the more liberal end, this estimate reached \$1,706,803,000.”

Reviewer #3 (Remarks to the Author):

****Synopsis****

In this manuscript Milham and colleagues present a descriptive analysis of papers that have used shared brain imaging data. They highlight where research has been published – specifically looking at the impact factor of each journal – along with the number of citations that each paper has received. They show increasing use of the INDI datasets, and that this increase is mostly driven by non-contributors (people who did not add data to the cohort used). The paper clearly describes some of the common “selfish” concerns about sharing data and provides evidence that they are largely unwarranted.

****Compliments****

I enjoyed reading this paper and I’m glad to see some analyses that can be used to respond to the “formidable – albeit hypothetical – downsides” of data sharing that often dominate conversations around sharing data. I found the paper clear and focused.

In particular, I really liked figure 3 which shows the additional benefit to the scientific literature that sharing large datasets provides. Researchers have a lot of ideas, but there are only 24 hours in the day. The number of publications by INDI contributors is increasing modestly, but the number using the data are much larger.

Thank you for sharing the code to re-create the dataset.

****Concerns/Suggestions****

HCP analyses

REVIEWER:

I’m concerned about the analysis of the publications from the Human Connectome Project on two fronts. The first is that the analyses as they stand rather undermine the main thesis of the paper, that is that using shared data – in particular “other people’s data” has no bearing on where research will be published. The blue line of figure 2D shows there are clearly more high impact publications by the HCP investigators than by non-investigators.

The authors do address this point in their results section: “It is worth noting that the publication impact pattern for papers authored by HCP investigators did appear to have an advantage over those published by non-investigators using the same data.” Two of the three possible interpretations they provide are common reasons why researchers would *not* want to share their data: protected lead time and an advantage understanding the data. (Why make it easy for others to reuse when you can keep publishing *and* meet your sharing requirements!)

I personally suspect that the different search strategy has a very large effect on the dataset presented. There are certainly more than 175 publications that have used HCP data, and I suspect that many of them are a) published in lower impact journals than the 28 major publications from the HCP investigators, and b) have HCP investigators named on them. My suspicion is that the papers in PubMed that include the term “Human Connectome Project” are a non-representative sample of the actual papers that use HCP data.

I’m not sure exactly what I recommend on this front, I understand from the methods section that it was a lot of work for the 5 research assistants to track all the publications using INDI datasets,

so it may not be possible to find all the publications that have used HCP data. I would be interested to hear the authors' response to my concern though, as they may be able to justify the sampling strategy better than is currently described in the paper.

Eitherway, I think it would be helpful to the reader to expand the discussion of the discrepancy between the HCP investigators and the INDI contributors. These two initiatives are very different in their approach to sharing and while I agree that the non-contributor users generalise nicely, the benefits to the organisers may be worth drawing attention as many of the readers will find (or have already found) themselves on the receiving end of a "data sharing mandate" from their funding body and there are major decisions to make regarding how accessible to make the data (even when officially available).

RESPONSE: We appreciate the reviewer's points. The knowledge added by the separation of HCP investigators and non-HCP investigators for this analysis needs to be weighed against the risk of inaccuracy arising from the limitations and biases introduced by the PubMed-based search. After careful consideration of this, we have opted to follow the suggestion of Reviewer 1 and err on the side caution, removing this aspect of the analysis. While as Reviewer 3 notes potentially intriguing statements may arise from this comparison, we cannot perform the analysis in an unbiased manner using PubMed and the work that would be required to perform it using Google Scholar would be excessive given that it is not a main focus of the manuscript.

Statistics

This isn't a big deal, but there are no statistics performed in the paper. I don't think there necessarily need to be, but I wonder if a sentence acknowledging that this is a descriptive study could be added to the methods section to make this clear.

RESPONSE: We have added the following sentence to our Methods section: "The work reported here is largely descriptive in nature; as such we do not include formal statistical analyses."

REVIEWER:

Comparative indices of impact

As I was reading the paragraph on publication impact in the results section (line 125) I wondered how these citation and impact metrics would compare to a randomly selected comparative sample. For example, I might hypothesise that these studies are more highly cited than others due to the larger (on average) sample sizes. Comparing these measures for the non-ABIDE studies (N=1834) or for the HCP studies (N=175) could be an easy analysis, but I'd be most interested to know how the measures compare to a sample of all human neuroimaging studies.

RESPONSE: In response to the reviewer's suggestion, we obtained a random sample (n = 4,000) of papers on Web of Science containing the keywords "MRI" and "brain" within the period from 2010 to 2017 (each year was equally represented within the sample). These samples were used to estimate the distribution of publication impact measures across all relevant human neuroimaging studies, finding results that were roughly consistent with those for INDI and the HCP. We have updated the manuscript as follows:

“Finally, we opted to extend these findings to what would be obtained for a sampling of the entirety of the MRI imaging literature for the same time period as INDI. To accomplish this, we obtained a random sample (n = 4,000) of papers on Web of Science containing the keywords “MRI” and “brain” within the period from 2010 to 2017 (each year was equally represented within the sample). The resulting density plot is similar to those obtained for INDI and HCP (the curves are higher for INDI and HCP over the general literature due to the greater number of publications in the higher-impact journals NeuroImage and Human Brain Mapping) (see Figure 2e).”

REVIEWER:

Informed consent

The paper does not address one of the major barriers to sharing data, that of the requirement to obtain informed consent before data can be made accessible to users. I don't think this manuscript needs to also add in a lot of detail about this barrier, but I wonder if it could be acknowledged somewhere in the paper to make this a slightly more complete resource for readers.

RESPONSE: We appreciate the reviewer's suggestion, and have added the following text to the Discussion in response:

“Fourth, there needs to be increased attention given to notification of the possibility of data sharing in the informed consent process. For studies designed with the intent to share data, whether due to their own desire or funding agency mandates, it is a given that the informed consent should make clear that the data will be shared and provide details as to the nature of sharing. However, for investigators who are not committed to data sharing, a key question is whether data sharing language should still be included in the informed consent. This is especially important given that incentives for sharing may arise over time (e.g., data sharing supplements for funded grants, data sharing consortia relevant to the investigator's interests).”

REVIEWER:

Measuring impact

The authors mention that “...the assessment of journal impact remains somewhat controversial

given the growing number of indices available (e.g., impact factor, CiteScore, altmetrics)” but I’d like to point out that it is not simple the number of indices that make JIF a controversial measure. A good discussion of their limitations can be found in Lariviere et al, BioRxiv, 2016, <https://doi.org/10.1101/062109>. I’d like to see more acknowledgement of these limitations in the introduction, along with an acknowledgement that they *are* used to assess tenure, promotion, job offers etc (and therefore the current study is worth conducting).

As a minor note, please describe how the h and i10 indices are calculated, as these terms are not defined in the text.

RESPONSE: We appreciate the reviewer’s points and have revised our text to acknowledge the limitations of these indices as follows:

“To address questions about whether high-impact journals accept secondary analyses of open data, we also examined journal impact factors for publications using INDI data. The assessment of journal impact remains somewhat challenging given the growing number of indices available (e.g., impact factor, CiteScore, altmetrics)²⁰, concerns about the appropriateness of judging the impact of individual articles based on measures of the impact of journals, and the potential over-reliance on these indices for promotion and tenure decisions. Nonetheless, for the purpose of citation analysis, they can be effective tools for summarizing literature-level trends.”

Additionally, we now include definitions for the h and i10 indices as follows:

“The impact of each of the major INDI efforts (FCP, NKI-RS, ADHD-200, ABIDE, CoRR) on the scientific literature was quantified using an array of commonly used citation-based indices, including the h- (the number of publications with at least the same number of citations) and i10- (the number of publications with at least 10 citations) indices (see Table 1).”

REVIEWER:

Data sharing

As I mentioned above, thank you for sharing the code required for the analyses. As the data will change between when the study was conducted and when the reader learns of it, it would be nice to have an archive of the dataset used to make the plots/tables in the manuscript, eg by uploading to an institutional repository or zenodo.

RESPONSE: In accord with the reviewer’s suggestion, we now include the data with the scripts on GitHub.

Kirstie Whitaker

REVIEWERS' COMMENTS:

Reviewer #1 (Remarks to the Author):

The revision properly addressed my original questions/suggestions and I do not have new ones. I fully support publication of this manuscript as it conveys an important message to the scientific community.

Dardo Tomasi

Reviewer #2 (Remarks to the Author):

Well done with the paper and the revisions - great work!

Reviewer #3 (Remarks to the Author):

I am delighted with the manuscript. The authors have addressed the constructive comments from all the reviewers very well and I look forward to citing this paper often. Thank you, and well done.

Kirstie Whitaker